# Digestive Well-Differentiated Grade 3 Neuroendocrine Tumors: Current Management and Future Directions

**DOI:** 10.3390/cancers13102448

**Published:** 2021-05-18

**Authors:** Anna Pellat, Anne Ségolène Cottereau, Lola-Jade Palmieri, Philippe Soyer, Ugo Marchese, Catherine Brezault, Romain Coriat

**Affiliations:** 1Department of Gastroenterology and Digestive Oncology, Hôpital Cochin, AP-HP, 27 rue du Faubourg Saint Jacques, Université de Paris, 75014 Paris, France; lolajade.palmieri@aphp.fr (L.-J.P.); catherine.brezault@aphp.fr (C.B.); romain.coriat@aphp.fr (R.C.); 2Department of Nuclear Medicine, Hôpital Cochin, AP-HP, 27 rue du Faubourg Saint Jacques, Université de Paris, 75014 Paris, France; annesegolene.cottereau@aphp.fr; 3Department of Radiology, Hôpital Cochin, AP-HP, 27 rue du Faubourg Saint Jacques, Université de Paris, 75014 Paris, France; philippe.soyer@aphp.fr; 4Department of Surgery, Hôpital Cochin, AP-HP, 27 rue du Faubourg Saint Jacques, Université de Paris, 75014 Paris, France; ugo.marchese@aphp.fr

**Keywords:** neuroendocrine neoplasms, well-differentiated grade-3, functional imaging, chemotherapy

## Abstract

**Simple Summary:**

We herein present the latest data on characteristics and therapeutic management of gastro-entero-pancreatic well-differentiated neuroendocrine grade 3 tumors (GEP-NET G-3). Although neuroendocrine neoplasms (NEN) of the digestive tract are rare tumors, progress in diagnosis has allowed their better identification. Since 2017, a new category of NEN showing well-differentiated morphology and high Ki-67 index has been defined, the NET G-3. These tumors show differences not only in morphology but also in genotype, clinical features, and treatment response, compared with classical high-grade NEN. Therefore, they should be managed differently but suffer from lack of data. We find our work important, underlying the need to conduct new rigorous clinical studies in this population.

**Abstract:**

Digestive well-differentiated grade 3 neuroendocrine tumors (NET G-3) have been clearly defined since the 2017 World Health Organization classification. They are still a rare category lacking specific data and standardized management. Their distinction from other types of neuroendocrine neoplasms (NEN) not only lies in morphology but also in genotype, aggressiveness, functional imaging uptake, and treatment response. Most of the available data comes from pancreatic series, which is the most frequent tumor site for this entity. In the non-metastatic setting, surgical resection is recommended, irrespective of grade and tumor site. For metastatic NET G-3, chemotherapy is the main first-line treatment with temozolomide-based regimen showing more efficacy than platinum-based regimen, especially when Ki-67 index <55%. Targeted therapies, such as sunitinib and everolimus, have also shown some positive therapeutic efficacy in small samples of patients. Functional imaging plays a key role for detection but also treatment selection. In the second or further-line setting, peptide receptor radionuclide therapy has shown promising response rates in high-grade NEN. Finally, immunotherapy is currently investigated as a new therapeutic approach with trials still ongoing. More data will come with future work now focusing on this specific subgroup. The aim of this review is to summarize the current data on digestive NET G-3 and explore future directions for their management.

## 1. Introduction

Neuroendocrine neoplasms (NEN) are most often located in the lung and in the digestive tract. They are defined by the expression of specific biomarkers, such as synaptophysin and chromogranin A (CGA), which can be absent in high-grade NEN [1,2,3,4]. The World Health Organization (WHO) grading system classifies neuroendocrine neoplasms (NEN) according to tumor differentiation and tumor grade (proliferation rate). Well-differentiated grade 3 neuroendocrine tumors (NET G-3) have been introduced in the 2017 WHO classification for pancreatic lesions; this was later generalized to all digestive tumor sites [5]. NET G-3 show well-differentiated morphology and high Ki-67 index (>20%) [2,6,7,8,9]. Both the proliferation rate (Ki-67 index and mitotic index) and differentiation status now separate NEN between well-differentiated NET (NET G-1 to G-3), and poorly differentiated neuroendocrine carcinomas (NEC) of the digestive tract (only G-3) [10] (Table 1).

Prior to this, all G-3 neuroendocrine samples, or high-grade NEN, were considered as NEC and uniformly evaluated and treated in studies [11]. Data on NEC is detailed in another paper of this special issue. It is now established that well-differentiated NET G-3 have a better prognosis than NEC because of distinct tumor characteristics [7,12,13]. Additionally, functional imaging has offered a new tailored approach in NEN detection, classification and treatment. With all this in mind, NET G-3 should benefit from a specific therapeutic approach with more trials expected to help define their precise management [14,15,16]. 

We present in this work the current data on clinical, radiological and pathological presentation as well as treatment, of digestive NET G-3, with a special focus on gastro-entero-pancreatic tumors (GEP-NET G-3).

## 2. Epidemiological Features and Tumor Presentation

### 2.1. Tumor Location and Incidence

Digestive NEN are rare tumors with a rising incidence in the context of diagnostic improvement [12,17]. High-grade NEN represent only a small sample of these tumors [18,19,20]. Precise data is still scarce for NET G-3 due to their recent identification: most data are retrospective and often come from reassessment and reclassification of NEC samples [6,7,8,12,21]. 

Studies show that NET G-3 are more often found in the pancreas with a frequency ranging from 10% to 65% [3,6,21,22]. Other main tumor sites are the colon/rectum and stomach, with frequencies ranging from 8% to 24% and 8% to 29%, respectively [3,22].

Regarding incidence, a 2015 retrospective study of 204 patients with GEP-NEN G-3 found 37 (18%) patients with NET G-3 and 167 (79%) with NEC [12]. Recently, a Korean study found 8 (10%) pancreatic NET G-3 (PanNET G-3) after reclassification of 82 surgically resected PanNEN [21]. One of the rare prospective studies, the PRONET study, included 1340 samples (biopsy, surgical, endoscopic resection and cytological preparations) of NEN and found 778 patients with GEP-NEN, including 104 (13.5%) NEN G-3. From those, the proportions of NEC, NET G-3 and mixed neuroendocrine–nonneuroendocrine neoplasm (MiNEN) were 69% (*n* = 72), 20% (*n* = 21) and 11% (*n* = 11) respectively [22,23]. In summary, the incidence of NET G-3 is probably underestimated, but seems to account for 20% of the NEN G-3 population. With their recent identification, future large epidemiological databases will probably help determine their actual incidence. 

### 2.2. Clinical Presentation and Biomarkers

Clinical presentation of digestive NEN shows great heterogeneity. Low-grade NETs are rather indolent tumors often associated with hormonal syndromes (i.e., functional tumors) and sometimes with a hereditary background. Most functional NET are found in the midgut, and functional PanNET represent only a minority of all PanNET [24,25]. On the opposite, NEC are aggressive neoplasms, rarely functional, and with unknown risk factors [3,7,20]. According to small available series, NET G-3 patients are more likely to have a functional tumor (5–50%) compared to NEC patients (0–6%) (Table 2) [3,6,7,26]. Various works found no significant differences in age compared with NET G-2 or NEC patients [4,6,7,21,27] with only two studies suggesting that patients with NET G-3 were younger than those with NEC at the time of diagnosis [3,26].

Specific biomarkers such as plasma CGA, plasma neuron specific enolase (NSE) and urinary 5-hydroxyindoleacetic acid (5-HIAA) are frequently used in NEN management. Plasma CGA helps monitor evolution and treatment response in well-differentiated NET [28,29,30] whereas NSE is more frequently assessed in high-grade NEN [31,32]. Urinary 5-HIAA is specific for carcinoid syndrome monitoring in midgut NET [33]. Data on plasma biomarkers are still very scarce for high-grade NEN. One study on 12 pulmonary and digestive NET G-3 found increased levels of plasma CGA and NSE or urinary 5 HIAA in 42%, 25% and 25% of patients respectively, with no significant differences with NEC patients [6]. Recently, another study found that blood concentration of FAS ligand (FASLG) was lower in 42 PanNEN G-3 compared to controls, and that positive FASLG immunoreactivity in tumor cells correlated with well-differentiated morphology (14/42 evaluable samples) [34]; suggesting that FASLG could be a future candidate biomarker. Finally, PanNET G-3 rarely show high levels of carbohydrate antigen (CA) 19–9 compared with pancreatic adenocarcinoma [35]. These preliminary results on biomarker assessments in NET G-3 need confirmation by future studies.

The scarcity of available data makes it hard to draw firm conclusions on NET G-3 clinical presentation. In case of diagnostic difficulty, a functional tumor is more in favor of NET G-3 than NEC, especially if located in the pancreas. There are currently not enough data to recommend the dosage of any plasma biomarker in digestive NET G-3.

### 2.3. Prognosis

The presence of metastasis is a well-known factor of poor prognosis in NEN, as for other neoplasms [13,36]. Both cell differentiation and proliferation rate are also major prognostic markers for NEN [2,9,12]. Grading, itself, plays a major role in the prognosis of well-differentiated NET, thus highlighting the importance of separating NET G-3 patients from others [13,37]. Indeed, a recent study in rectal NET, including four patients with NET G-3, showed that grade was significantly associated with distant metastasis [38]. 

In various works, the overall survival (OS) for NET G-3 patients was longer than for NEC patients: median survival ranged from 41–99 months versus (vs.) 5.3–17 months [3,6,7,26,39]. Regarding PanNEN, one study showed that OS for both NET G-2 and NET G-3 patients was similar (67.8 and 54.1 months, respectively; *p* = 0.02) and significantly higher than for NEC patients (11 months) [7]. Patients with PanNET G-3 also show better OS than those with ductal pancreatic adenocarcinoma [35]. These results indicate that NET G-3′s prognosis falls in between that of NET G-2 and NEC, and more data are needed for non-pancreatic lesions.

## 3. Imaging 

### 3.1. Functional Imaging

Functional imaging plays an important role in detection, classification as well as treatment of NEN [40]. Indeed, positron emission tomography/computed tomography (PET-CT) imaging can be used for the selection of candidate patients for peptide receptor radionuclide therapy (PRRT) but can also provide evidence of treatment response (see Section 5). ^18^F-Fluorodesoxyglucose (^18^F-FDG) PET-CT is recommended to help in NEC diagnosis and often indicates poor prognosis when positive in well-differentiated NEN [41,42,43]. Well-differentiated NET overexpress somatostatin receptors (SSTRs) and most of them show positive somatostatin receptor imaging (SRI) uptake (Octreoscan^TM^ or ^68^Ga-DOTATOC PET-CT). Furthermore, the combined use of SRI and ^18^F-FDG PET-CT has led to the development of the NETPET grading score, to help classify and select patients for individualized treatment (i.e., patients with SSTR- and ^18^F-FDG+ are less likely to respond to PRRT) [44,45]. Indeed, results with dual tracers were correlated with cell differentiation and Ki-67 index values [46].

In the small available series, most patients with NET G-3 showed positive SRI uptake (Table 2) [3,6,26]. In Velayoudom-Céphise et al. [6] and Heetfeld et al. [3] studies, patients with NET G-3 showed significantly more positive SRI than those with NEC (*p* = 0.03 and 0.001 respectively). In a recent work, six NET G-3 lesions showed positive ^68^Ga-DOTATOC PET-CT uptake, and a negative correlation was found between Ki-67 index and SUVmax values in the whole NET population evaluated (r = −0.3; *p* = 0.018) [47]. Regarding ^18^F-FDG PET-CT, data are even more scarce (Table 2). In Velayoudom-Céphise et al. work, 25% of NET G-3 patients had a positive ^18^F-FDG PET-CT uptake [6]. In Heetfeld et al. work, only 12 patients of the NET G-3 group were evaluated and 9 had a positive ^18^F-FDG PET-CT uptake (75% of positivity), which was similar with NEC patients [3]. These results indicate that NET G-3 are more likely to have a positive SRI and should benefit from this type of examination, but ^18^F-FDG PET-CT is not helpful for distinguishing between NET G-3 and NEC. Specific data are still lacking regarding the use of dual tracers in NET G-3, but some authors recommend that both SRI and ^18^F-FDG PET should indeed be performed given encouraging results with PRRT in this population [48]. 

We show here the results of dual tracer functional imaging (^68^Ga-DOTATOC PET-CT and ^18^F-FDG PET-CT) for one patient with metastatic PanNET G-3 presenting with positive SRI uptake and incomplete ^18^F-FDG PET-CT uptake (Figure 1).

Several preclinical studies have reported on C-X-C motif chemokine receptor 4 (CXCR4)-targeted molecular imaging and therapy. CXCR4 is overexpressed in GEP-NET and could act as a potential target for treatment [40]. Pentixafor, an analog of a CXCR4 ligand, has been approved by the Food and Drug Administration in 2008. Werner et al. have studied the feasibility of CXCR4-PET in 12 GEP-NET, including five NET G-3, using ^68^Ga-Pentifaxor in comparison with ^68^Ga-DOTATOC PET-CT and ^18^F-FDG PET-CT: 80% of NET G-3 exhibited positive CXCR4-PET uptake (^68^Ga-Pentifaxor positive lesions) and increasing receptor expression was observed with increasing tumor grade [49]. These results were based on a limited number of patients and need further confirmation, but CXCR4-targeted therapy might possibly play an important role in the future in the field of theranostics for NET G-3.

Regarding other types of functional imaging, there are very little data on PET-magnetic resonance imaging (MRI) in NEN, and none in NET G-3 to our knowledge [50].

### 3.2. Morphological Imaging and Radiomics

Apart from functional imaging, CT and MRI are also frequently used for NEN management in clinical practice, especially for treatment response evaluation. When it comes to tumor diagnosis, researchers have found that several morphologic features such as ill-defined margins, large tumor size, heterogeneous and poor to moderate enhancement, vascular involvement, upstream Wirsung duct dilatation, and distant metastases are less frequently observed in G-1/G-2 NET by comparison with G-3 NEN [51,52,53]. One recent work looked into the specificities of morphological imaging in 13 patients with NET G-3 compared to 23 with NEC [54]. Patients with poorly differentiated lesions showed larger sized tumors, more necrosis and lower attenuation on pre-contrast and on portal venous phase CT images, with all results being significant [54]. Hemorrhagic content on MRI was only present in NEC (*p* = 0.007) [54]. Due to the rarity of NEC and NET G-3, morphological criteria are mainly used in dedicated multidisciplinary meetings in centers with large output to predict invasiveness.

Radiomics refers to the high-throughput extraction and analysis of quantitative features from medical images [55]. Researchers have also shown that NET G-3 have a suggestive CT radiomics signature that helps differentiate them from NET G-1 and NET G-2 or from NEC [51,53,54,56]. 

We present here the CT and MRI images of a patient with PanNET G-3 to illustrate the previously described high-grade morphological features (Figure 2).

Overall, a functional high-grade NEN with positive SRI uptake is in favor of NET G-3 rather than NEC, especially when located in the pancreas. Morphological imaging can also be used by expert teams to help identify NET G-3 in case of diagnostic difficulty. 

Following these first two chapters of our review, we have summarized the results from the largest series studying NET G-3 patients (Table 2).

## 4. Histopathology and Molecular Biology

### 4.1. Morphology and Ki-67 Index

Pathological evaluation is crucial for high-grade NEN classification and often requires an expert NEN pathologist. High-grade NEN show differences in their morphologic features. In PanNET G-3, tumor cells are mostly cuboidal in shape, with abundant cytoplasm and uniform nuclei [21,27]. There is an increased proliferation accompanied with changes in morphology such as apoptosis, mitoses and nuclear tangles [27]. Also, compared with NEC, NET G-3 show less pleomorphism and necrosis [27]. The presence of another histological type can sometimes result in tumor diagnosis difficulty (mixed morphology). The 2017 WHO classification has introduced the notion of MiNEN where any other histological type can be associated with the neuroendocrine morphology (at least 30% of the tumor sample) (Table 1) [2,9].

Regarding the proliferation rate, Ki-67 index values usually range from 21 to 50% in NET G-3, whereas NEC often show higher results up to 100% (Table 2). Although no data on differentiation was available, the NORDIC study on 305 patients with GEP-NEN G-3 showed that patients with Ki-67 index <55% had better OS and different treatment response [57]. Following these results, some authors have suggested that the 55% Ki-67 value could be the best cutoff to distinguish well-differentiated NEN G-3 from NEC [58]. To this day, this has not been validated and clinicians should follow the WHO guidelines for NEN G-3 classification. 

An accurate pathological assessment of the Ki-67 proliferation is essential for NEN diagnosis. Technical factors such as the specimen type (biopsy, needle aspiration cytology or surgical specimen), the staining technique, the choice of antibody and the assessment method may potentially affect the reproducibility of Ki-67 index values [59]. Recently, Kalantri et al. found that grading of PanNEN on cytology samples collected by endoscopic ultrasound-guided fine-needle aspiration showed good agreement with results from histology samples [60]. Manual counting (MC) of >2000 cells is considered the “gold standard” method by the WHO grading system. MC and digital image analysis (DIA) appear more reliable than “eyeballing” because of marked interobserver and intra-observer variability [61,62]. This was particularly observed for the assessment of tumor grades (G-1/G-2 and G-2/G-3 cutoffs) [61]. Nevertheless, in another work Ki-67 index assessment by “eyeballing” was highly correlated with results in DIA [63]. Additionally, compared with DIA, MC and “eyeballing” tended to overestimate the Ki-67 index [61,63]. In practice, MC remains the gold standard for evaluation of Ki-67 index and can be performed through the microscope or on screenshot printed image, which seems the most practical method based on its cost/benefit ratio and reproducibility [62].

Evolution of a well-differentiated NET to a high-grade NEN has been suggested by some authors. In a single-center retrospective study on 46 patients with 106 lesions of PanNET, increase in tumor grade occurred in 28 patients (63.6%) with the majority evolving from G-1/G-2 to G-3 [64]. On top of that, high progression correlated with worst survival [64]. In total, possible grade evolution remains a hypothesis. Other explanations such as sampling error or mixed tumors could also be considered in case of a patient’s unexpected clinical presentation. Therefore, multiple biopsies in one patient can sometimes be performed to guide future treatment.

### 4.2. Molecular Biology 

Even if pathological diagnosis has improved, in rare cases a grey zone persists for pathologists to distinguish NEC from highly proliferative NET G-3. Morphological characterization can be difficult in tumors with important tumoral heterogeneity and/or necrosis [27]. Here, molecular biology findings have greatly helped in the distinction between these two entities. NET G-3 seem to have common pathogenetic mechanisms with low and intermediate grades NEN, with key drivers of their molecular pathogenesis differing from those in NEC. As mentioned above, some researchers have suggested that PanNET G-3 might develop from an initial G-1 or G-2, whereas PanNEC could develop from ductal adenocarcinoma [61,65]. Small and large-cell NEC of the digestive tract show genetic similarities with frequent inactivation of the TP53, Rb and SMAD4 pathways, due to intragenic mutations in the *TP53*, *RB1* and *SMAD4* genes [66,67,68,69]. These genetic changes are rarely seen in well-differentiated PanNEN [66,67]. Conversely, inactivating mutations in *DAXX* and *ATRX* and in *MEN1* are found exclusively in PanNET [68,70,71]. Mutations in other components of the PI3 K/mTOR signaling pathway including *PTEN*, *DEPDC5*, and *PIK3CA* have also been observed in PanNET [66,70,72]. Finally, a whole-genome sequencing of liver metastasis in a patient with metastatic PanNET G-3 exhibited a *TSC1*-disrupting fusion, a novel *CHD7–BEND2* fusion, but lacked any somatic variants in *ATRX*, *DAXX*, and *MEN1* [73]. Following these results, in 2016 Tang et al. had proposed a diagnostic algorithm based on frequent molecular alterations to help distinguish between PanNET G-3 and PanNEC [67]. More data is urgently needed in non-pancreatic NET G-3. 

We have summarized the main molecular alterations found in high-grade NEN in Table 3.

## 5. Treatment 

Therapeutic management of digestive NET G-3 is not completely standardized because of their rarity and the lack of well conducted robust trials. Here, we address the different therapeutic options with available data in NET G-3.

### 5.1. Surgery and Liver-Directed Therapies 

Until recently, most surgical series of NEN G-3 have included a heterogeneous population of patients with well and poorly differentiated tumors, making it hard to draw firm conclusions. Despite this issue, European and American guidelines recommend surgical resection irrespective of tumor grade and differentiation in the non-metastatic setting [74,75,76,77,78,79,80]. Surgery can also be performed in well-differentiated NET G-1 and G-2 after neoadjuvant therapeutic approach in patients with important initial tumor burden (locally advanced tumor or large resection needed), or in the presence of metastasis. The overall prognosis is mainly based on tumor burden (especially in the presence of liver metastasis), cell differentiation and proliferation rate, but also initial tumor site (risk of occlusion in midgut NET).

One study on 28 patients operated for PanNEN G-3 suggested that patients with NET G-3 have similar postoperative survival compared to those with NEC, which was significantly lower than for NET G-2 and G-1 patients [81]. This is contradictory with current knowledge on NET G-3 prognosis and these results could be explained by the pooled evaluation of both localized and metastatic tumors [3,26,67,82]. Although surgery is recommended in the non-metastatic setting, we need to keep in mind that high-grade is associated with higher risk of recurrence and disease specific-death, as shown recently on a series of operated PanNEN [83]. 

Liver-directed therapies can be performed alone or in combination with surgery. Indeed, they have shown good clinical and morphological responses for well-differentiated G-1 and G-2 NET, especially when liver burden is important or in the presence of a secretory syndrome. To date, there is no specific data for this therapeutic approach in NET G-3 [84,85]. 

Although data on surgery in NET G-3 are scarce, it is still considered as the first valid option in the localized setting for all well-differentiated NET. It should be individually discussed for metastatic NET G-3 patients in the context of other available therapeutic approaches such as chemotherapy. Liver-directed therapies should also be discussed individually and in case of important secretory syndrome.

### 5.2. Somatostatine Analogues and Targeted Therapies

Both PROMID and CLARINET prospective trials have validated the anti-proliferative effect of somatostatine analogues (SST) in GEP-NET G-1 and G-2 [86,87,88]. They are mainly used for indolent well-differentiated NET in the first-line setting and for treatment of the secretory syndrome [78,79]. Both studies also showed that SST have a higher efficacy in tumors with low Ki-67 index, low hepatic load and slow pretreatment growth [50,51,52]. There were no G-3 lesions included in these trials so the use of SST in this population should be limited and only considered with a close monitoring for its effect on the secretory syndrome. 

Targeted therapies are mainly used for the treatment of advanced well-differentiated NET. The European Neuroendocrine Tumors Society (ENETS) guidelines recommend them in both first or second-line settings when chemotherapy is not appropriate [89]. In advanced PanNET G-1 and G-2, sunitinib [90], a tyrosine kinase inhibitor, and everolimus [91], a mTOR inhibitor, have both demonstrated efficacy in randomized phase III studies. Regarding advanced non-pancreatic NET, everolimus has shown some efficacy in the RADIANT 2 trial with results later confirmed by the RADIANT 4 trial [92,93]. One work on a small sample of high-grade NEN, including at least six NET G-3 patients, has shown evidence of sunitinib activity [94]. Four out of six patients had partial response or stabilization of the disease under treatment [94]. In one study of 15 patients with “well-moderately” differentiated PanNEN G-3 tumors, administration of everolimus as first-line treatment showed sustained disease stabilization for three out of four patients [95]. The EVINEC phase II trial has evaluated the safety and tolerability of everolimus as second-line treatment in NEC and NET G-3, but results are not yet available (NTC02113800). Recently, surufatinib, a novel multi-target inhibitor, has shown positive effects in progression-free survival (PFS) for extra-pancreatic NET of low and intermediate grades vs. placebo, with no data in NET G-3 [96]. Overall, results with everolimus and sunitinib in NET G-3 need confirmation in larger populations and cannot currently be proposed as first-line treatment. 

### 5.3. Chemotherapy

Chemotherapy is a key treatment in metastatic PanNET irrespective of grade. It is commonly prescribed in the first-line setting for “aggressive” metastatic PanNET or in case of failure of previous therapies. Multiple combinations of chemotherapy regimen have been validated in G-1 and G-2 PanNET such as streptozotocin/doxorubicin [97], 5-fluorouracil/streptozotocin [97], LV5 FU2/dacarbazine [98] and capecitabine/temozolomide [99,100]. The addition of bevacizumab to a 5-fluorouracil/streptozotocin combination showed a significant disease control rate (DCR) (56% of partial responses and 44% of stabilizations) in the BETTER trial [101]. Irinotecan with 5-fluorouracil (FOLFIRI) can also be been considered as an option in second-line treatment of PanNET [102]. Finally, other combinations of chemotherapies such as capecitabine and oxaliplatine (XELOX), or gemcitabine and oxaliplatine (GEMOX) have also shown effective results in PanNET [103,104].

In non-pancreatic metastatic NET, there is no standard of care regarding chemotherapy [78,79]. Various regimens have been evaluated with studies showing low response rates, especially with alkylant-based treatments [105,106,107]. This could be the result of the strong expression of O_6_-methylguanine DNA methyltransferase (MGMT) in these tumors [108]. Cassier et al. suggested that GEMOX had some efficacy in pre-treated non-pancreatic NET G-1 and G-2 patients, with an 84% overall response rate (ORR) [109]. The best results regarding chemotherapy were obtained with the BETTER trial evaluating the capecitabine/bevacizumab combination: in 49 chemotherapy-naïve patients treated, there was an ORR of 88% and a PFS of about 23 months [110].

In metastatic NEC from all sites, the recommended first-line chemotherapy is the combination of platinum derivatives (cisplatin or carboplatine) with etoposide [111,112]. A combination of irinotecan and cisplatin can also be proposed, following the results of a Japanese study [113]. FOLFIRI or the association of 5-fluorouracil and oxaliplatine (FOLFOX) can be administered in the second-line setting [114,115]. One study has also suggested some efficacy of second-line temozolomide-based regimen in digestive NEC, with 71% of response (partial response or stabilization). A trial is currently ongoing comparing platinum-based chemotherapy to temozolomide-based chemotherapy in the NEC population (NCT02595424) [116].

In metastatic NET G-3 the efficacy of platinum-based chemotherapy seems limited, with response rates ranging from 0% to 10%. Two studies reported both 0% objective response rate to platinum-based chemotherapy in their NET G-3 patients [4,6]. Similarly, in Heetfeld et al. study, the response rate to platinum-based chemotherapy was 2% in NET G-3 patients vs. 39% in NEC patients [3]. In another series of 16 patients with PanNET G-3 the response rate to platinum agents was 10% [26]. In PanNEN G-3 patients from the NORDIC study there was a higher response to platinum-based regimen for patients with Ki-67 index >55% (42% vs. 15%), suggesting that aggressive lesions respond better to this regimen [57]. On the opposite, results with alkylant agents are more encouraging in NET G-3 patients. In the 16 patients with PanNET G-3 from Raj et al. study, the response rate with alkylant-based chemotherapy was 50% [26]. Furthermore, a monocentric study on G-2 and G-3 NEN from various sites (including 11 NET G-3 patients) evaluated the effect of capecitabine-temozolomide (CAPTEM) with 22% patients treated in the first line setting [117]. There was a trend towards improved median PFS in patients with NET G-3 and Ki-67 index <55% (15 vs. 4 months, *p* = 0.117) and for patients who received CAPTEM as first-line therapy (17 months vs. 8 months, *p* = 0.3) [117]. Toxicity with CAPTEM regimen was manageable [117,118]. Similarly, other studies have found positive results with the CAPTEM regimen in small samples of NET G-3 [119,120,121].

All of these results suggest that the choice of chemotherapy regimen in metastatic PanNET G-3 should be in line with NET G-2 rather than NEC, especially when Ki-67 index <55%. Platinum-based regimen can be proposed after individual discussion only. In non-pancreatic NET G-3, no chemotherapy regimen should be considered as a standard of first-line care considering the very small amount of data available.

### 5.4. Peptide Receptor Radionuclide Therapy (PRRT)

The efficacy of PRRT (Lutetium−177 (^177^Lu)-Dotatate) combined with SST was proven in the prospective phase III NETTER-1 study in 229 patients with advanced well-differentiated midgut NET [122]. The study showed a benefit in PFS at 20 months and in response rate, compared with the control group [122]. Positive results with PRRT are also found in PanNET G-1 and G-2 [123]. Furthermore, a phase I study assessing the efficacy and safety of a novel SST antagonist (^177^Lu)-satoreotide tetraxetan has shown promising results in 20 patients with well-differentiated NET (including one patient with NET G-3) [124].

As previously described, NET G-3 patients are more likely to show positive SRI uptake than NEC patients. PRRT has been proposed for high-grade NEN patients showing foci anatomical agreement of the SRI and glucose uptake lesions (little mismatch). Recent data are in favor of PRRT efficacy in the second or third-line settings for high-grade GEP-NEN. A review of four studies with at least 10 NEN G-3 patients treated with PRRT showed promising response rates (31–41%) and disease control rates (69–78%) in this population [125]. Both PFS (11–16 months) and OS (22–46 months) were best for patients with Ki-67 index <55% [125]. Even if there were some differences between the four considered studies, they all showed that about two thirds of the pooled population of NEN G-3 had a potential to respond to PRRT. In Carlsen et al. study in 149 NEN G-3 patients treated with PRRT, including at least 60 NET G-3 patients, response rates did not differ among subgroups (including differentiation) [126]. Median PFS and OS were significantly longer for patients with Ki-67 index <55% and well-differentiated tumors (*p* < 0.001) [126]. Recently, the combination of PRRT and chemotherapy has been evaluated in advanced G-2 and G-3 NEN. One study found that CAPTEM combined with PRRT had significant activity with mild toxicities in a population including eight NEN G-3 and showing positive dual tracer expression [121].

These results suggest that, for carefully selected patients, PRRT should be considered after first-line treatment for both NET G-3 and NEC with increased uptake on SRI and little mismatch, especially when Ki-67 index <55% [127]. In this sense, dual tracer using ^18^F-FDG PET-CT and SRI can provide important information for NET G-3 selection for PRRT [125,128]. 

### 5.5. Immunotherapy

As for other types of malignancies, immune checkpoint inhibitors (ICI) have been considered in NEN treatment. The KEYNOTE-158 trial showed that pembrolizumab had limited antitumor activity in previously treated advanced NET G-1 and G-2 from various sites, with an ORR of 3.7% (95%CI, 1.0–9.3) [129]. Some case reports have described treatment response or long survivals in high-grade NEN patients treated with ICI, suggesting that they are a more promising treatment option in this population [130]. For instance, programmed-death-1 blockage has shown positive results in both first and second-line treatment of Merkel cell carcinoma, a high-grade cutaneous NEC [131,132]. Efficacy of ICI in high-grade NEN could be explained by microsatellite instability, and/or high mutational load, which are more frequent in these tumors. Nevertheless, recent data showed that pembrolizumab alone had limited effect in NEN G-3 patients, with a DCR of 24.1% [133]. Several phase II studies investigating the effect of ICI in patients with advanced high-grade NEN, including NET G-3, are still ongoing. Avelumab is currently evaluated in advanced well-differentiated NET G-2/G-3 (NCT03278379) as well as in progressive NEC/NET G-3 after chemotherapy (NCT03352934). Finally, the combination of durvalumab and tremelimumab is being investigated in GEP-NEN G-3 after progression to previous therapies (NCT03095274). Further studies might also investigate the combination of ICI and chemotherapy or other treatment in the therapeutic management of NEN G-3. Up to this day, ICI for NET G-3 can only be proposed in clinical trials.

## 6. Conclusions and Perspectives

NET G-3 are rare tumors showing specific features of clinical interest. Their prognosis seems closer to that of NET G-2 rather than that of NEC, but with a worse OS. If in doubt, pathologic reassessment by a NEN expert should be easily proposed, especially for pancreatic lesions where NET G-3 more often arise. Due to their rarity and the specific management they require, NET G-3 treatment should always be discussed in NEN expert meetings. Functional imaging plays a crucial role in both diagnosis and treatment management of NEN, and dual tracer imaging should easily be proposed for NET G-3 patients. To this date, available treatments include mainly surgery and chemotherapy with alkylant-based regimen and sometimes platinum-based regimen (most of the data coming from pancreatic series). There are promising results with targeted therapies and PRRT which need confirmation in larger populations. Trials with ICI are still ongoing, and the combination of chemotherapy and ICI should be explored in the future for more antitumoral activity. Based on the available literature and on the 2020 European recommendations, we propose the following algorithm for the therapeutic management of digestive NET G-3 [127] (Figure 3).

Currently, molecular biology and theranostics seem the two most promising fields of research to help individualize and standardize treatment for NET G-3. Whole genomic sequencing of PanNET G-3 samples could help identify new structural rearrangements or mutations. Data in molecular biology is also urgently needed in non-pancreatic NET G-3. Other diagnostic tools, such as the NET test, a multianalyte liquid biopsy measuring NET gene expression, might also help individualize NET G-3 in the future. Finally, dual tracer imaging should always be discussed in patients with NET G-3 to help select patients for PRRT. 

Overall, it is important to highlight that our review has limitations due to the scarcity of available studies for this newly introduced entity. Additionally, the population samples are very small. Therefore, the main conclusions on NET G-3 characteristics and specific treatment will need to be confirmed in future larger trials. In order to deal with recruitment difficulty due to the rarity of these tumors, international collaborations should be encouraged.

## Figures and Tables

**Figure 1 cancers-13-02448-f001:**
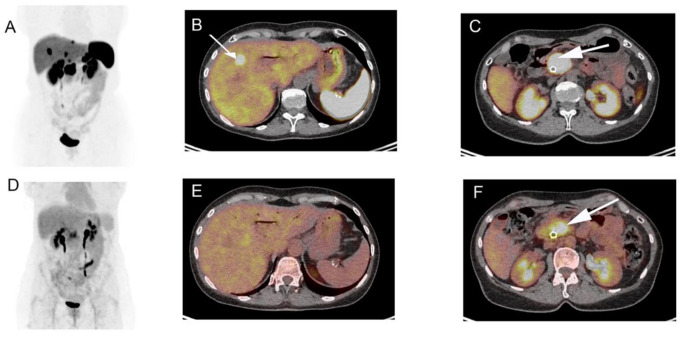
A 45-year-old woman presenting with a large lesion of the pancreatic head, with an initial diagnosis of grade 2 neuroendocrine tumor (NET G-2) after histopathologic analysis of biopsy specimens. Pre-therapeutic ^68^Ga-DOTATOC. positron emission tomography/computed tomography (PET-CT) maximum intensity projection image (**A**) and axial fused PET-CT images (**B**,**C**) showed multiple liver lesions (**B**, arrow) and high uptake by pancreatic lesion (**C**). The ^18^F-FDG PET-CT (**D**–**F**) demonstrated high focal uptake only in the central region of the pancreatic tumor (**F**) and no pathological liver uptake (**E**), highlighting tumor heterogeneity. Finally, histopathologic analysis of surgical specimens revealed a NET G-3 with Ki-67 index of 22% and confirmed liver metastasis.

**Figure 2 cancers-13-02448-f002:**
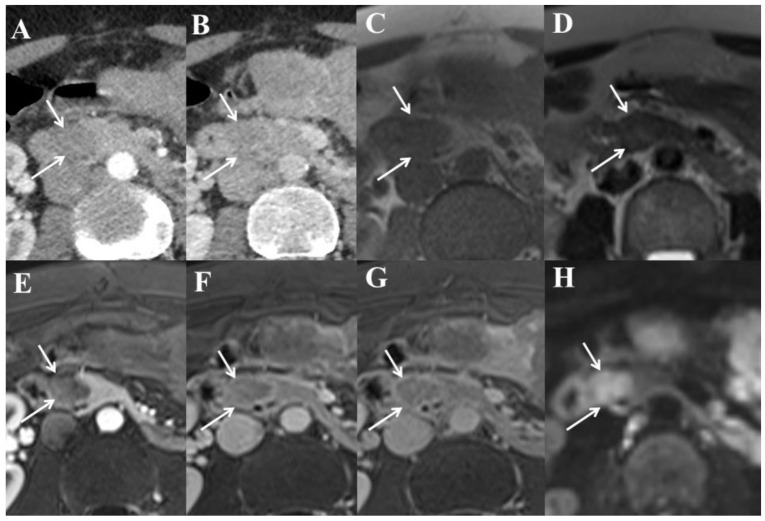
Forty-six-year-old woman with well-differentiated grade 3 neuroendocrine tumor (Ki-67 index = 28%) of the pancreatic head (PanNET G-3). (**A**) Computed tomography (CT) image in the axial plane obtained during the arterial phase following intravenous administration of iodinated contrast material reveals ill-defined, slightly hypoattenuating lesion (arrows) of the pancreatic head. The poor enhancement during arterial phase is in contrast with the classical hyperenhancement observed in well-differentiated PanNET G-1 and G-2. (**B**) On CT image obtained during the portal phase, the lesion (arrows) is hardly visible. (**C**) T1-weighted magnetic resonance (MR) image in the axial plane obtained before intravenous administration of a gadolinium-based contrast agent shows hypointense pancreatic lesion (arrows). (**D**) T2-weighted MR image shows ill-defined and slightly hyperintense pancreatic lesion (arrows). (**E**) T1-weighted MR image in the axial plane obtained during the arterial phase following intravenous administration of a gadolinium-based contrast agent clearly shows well-defined and markedly hypointense pancreatic lesion (arrows). The poor enhancement during arterial phase is in contrast with the classical hyperenhancement observed in well-differentiated PanNET G-1 and G-2. (**F**) On T1-weighted MR image obtained during the portal venous phase of enhancement, the lesion (arrows) remains hypointense relative to the adjacent pancreatic parenchyma. (**G**) On T1-weighted MR image obtained during the late phase of enhancement, the lesion (arrows) remains hypointense relative to the adjacent pancreatic parenchyma. (**H**) Diffusion-weighted MR image (b = 1000 s/mm^2^) in the axial plane shows hyperintense pancreatic lesion (arrows). The tumor had a low apparent diffusion coefficient of 0.952 × 10^−3^ mm^2^/s.

**Figure 3 cancers-13-02448-f003:**
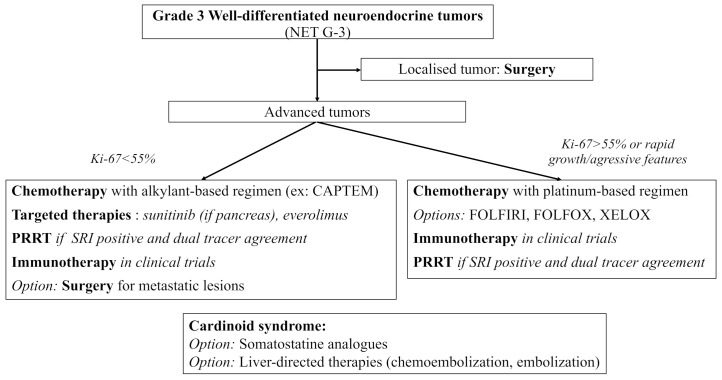
Proposed treatment algorithm for the therapeutic management of grade 3 well-differentiated neuroendocrine tumors (NET G-3). CAPTEM: capecitabine + temozolomide, FOLFIRI: 5-fluorouracile + irinotecan, FOLFOX: 5-fluorouracile + oxaliplatine, XELOX: capecitabine + oxaliplatine, PRRT: peptide receptor radionuclide therapy, SRI: somatostatin receptor imaging.

**Table 1 cancers-13-02448-t001:** The 2019 World Health Organization (WHO) classification for neuroendocrine neoplasms (NEN) of the digestive tract.

Well-Differentiated NEN ^1^	Ki-67 Index (%)	Mitotic Index (HPF ^2^/10 HPF)
NET ^3^ G-1 (low-grade)	<3	<2/10
NET G-2 (intermediate-grade)	3–20	2–20/10
NET G-3 (high-grade)	>20	>20/10
Poorly differentiated NEN		
NEC ^4^ G-3 Small-cell type, Large-cell type	>20	>20/10
Mixed neuroendocrine–nonneuroendocrine neoplasm (MiNEN)		

^1^ NEN: neuroendocrine neoplasm; ^2^ HPF: high-power field; ^3^ NET: neuroendocrine tumor; ^4^ NEC: neuroendocrine carcinoma.

**Table 2 cancers-13-02448-t002:** Results from main studies evaluating grade 3 neuroendocrine tumors (NET G-3) patients.

Study	Velayoudom-Céphise et al. [6] 2013	Heetfeld et al. [3] 2015	Basturk et al. [7] 2015	Scoazec et al. [23] 2017	Hijioka et al. [4] 2017	Raj et al. [26] 2017	Kim et al. [21] 2020
Sample size	12	37	19	21	21	16	8
Age (year)	56 *median*	52 *median*	54 *median*	NA ^7^	63 *median*	47 *mean*	57 *median*
Tumor location	Pancreas, non-digestive	Pancreas, rectum, stomach, small bowel	Pancreas	Pancreas, colon, rectum, stomach, small bowel	Pancreas	Pancreas	Pancreas
Metastatic state (%)	100	62	67	NA	71	69	100
Median Ki-67 (%)	21	30	40	35	29	47	23
Positive SRI ^1^ uptake N ^2^ of patients Type of imaging % of positivity	7/8 Octreoscan^®^ 88	21/24 Not specified 87.5	NA NA NA	NA NA NA	NA NA NA	13/15 Octreoscan^®^ 87	NA NA NA
Functional tumor	3 (25%)	5 (14%)	1 (5%)	NA	NA	8 (50%)	0 (100%)
ORR ^3^ (%)	NA	2/12 (17)	NA	NA	0/16 (0)	NA	NA
Median OS ^4^ (months)	41	99	54	NA	42	52	87
Median PFS ^5^ (months)	NA	NA	NA	NA	NA	NA	16
DCR ^6^ (%)	NA	3/12 (33)	NA	NA	6/16 (37,5)	NA	NA

^1^ SRI: somatostatin receptor imaging; ^2^ N: number; ^3^ ORR: overall response rate; ^4^ OS: overall survival; ^5^ PFS: progression free survival; ^6^ DCR: disease control rate, ^7^ NA: not available. Source: adapted and updated from Pellat et al. [15].

**Table 3 cancers-13-02448-t003:** Main molecular alterations found in grade 3 neuroendocrine tumors (NET G-3) versus neuroendocrine carcinoma (NEC).

Molecular Alterations (%)	NET G-3 ^1^	NEC ^2^
Rb1 mutation	0	67–75
KRAS mutation	0	28–50
p53 mutation	0	57–87
SMAD4 mutation	0	5
Loss of DAXX/ATRX expression	45–47	0

^1^ NET G-3: grade 3 neuroendocrine tumor; ^2^ NEC: neuroendocrine carcinoma.

## Data Availability

This narrative review is based on previously published data.

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
