# Peer review of "Digestive Well-Differentiated Grade 3 Neuroendocrine Tumors: Current Management and Future Directions"

_cancers, 2021, doi:10.3390/cancers13102448_

Round 1
Reviewer 1 Report
The Authors have substantially improved the manuscript and provide an excellent revision of their initial submission. My comments have been addressed adequately.
Author Response
Thank you very much
Reviewer 2 Report
The paper contains valuable information on an evolving topic. The authors are very focused on presenting data at the expense of presentation and flow of the paper. It is content dense, but difficult to follow in areas and often neglects appropriate introductory and summary statements that facilitate reader understanding. E.g 1) Immunotherapy - introductory sentence is the result of a clinical trial, 'The KEYNOTE-158 trial shows...', 2) Somatostatin analogues - last/concluding sentence 'the results from the EVINEC trial will probably shed some light in the second line setting'. There is excessive use of adverbs such as; 'indeed', 'nevertheless', 'to this day', etc. that are not standard in scientific writing. I would recommend having it edited by an experienced medical writer to ensure it is presented as succinctly as possible.
Author Response
Thank you for this valuable comment. We have done some new modifications to make the article easier for understanding. We have deleted many adverbs, added introductory sentences when needed, and changed the order of some sentences or paragraphs. Also, we have changed the order of figures or tables when needed. These changes were done under the supervision of all authors, most being experienced medical writers. Please find the new revised version of the manuscript attached.
This manuscript is a resubmission of an earlier submission. The following is a list of the peer review reports and author responses from that submission.
Round 1
Reviewer 1 Report
The Authors present a review on the rare tumor entity of WD G3 GEP-NETs with the aims to present the latest data on the disease characteristics and evaluate its therapeutic management. In my view, the manuscript could be of interest to the clinical NET community as it summarizes well valuable information; therefore, I believe it would be of interest to the readership of Cancers. However, there are some major points that need to be addressed that could improve the quality of the paper. The manuscript would benefit from major revision, visualization of existing evidence in comprehensive tables and figures and clarification with respect to its limitations.
Reviewer’s comments.
- Please discuss briefly the suggested pathophysiological background of G3 NETs including high grade progression and provide a figure on the the molecular pathogenesis of these tumors.
- Please comply to PRISMA guidelines and report a search strategy from pertinent databases for the studies included in this review.
- With regards to functional and morphological imaging/radiomics please provide a table summarizing the evidence presented in the included studies discussed in the relevant sections.
- Along the same line, in the context of dual functional imaging with 68Ga-DOTA-TOC/TATE/NOC-PET and 18F-FDG-PET scan , please specify that PRRT has been proposed in this scenario under the condition of foci anatomical agreement of the gallium and glucose uptake lesions.
5. Mestier et al 2020 Neuroendocrinology, and Thomas et al. 2020 Cancers, provide data on TEMCAP treatment also in the subset of NET G3. Please consider including these references in the relevant discussion part.
6. With regards to PRRT, please consider to briefly discuss the novel therapeutic scheme of combining PRRT and chemotherapy in parallel for patients whose dual functional imaging scans show increased SSTR density and high glucose uptake. Several trials on agents such as 5-fluorouracil (5- FU), capecitabine (CAP) and temozolomide (TEM) have recently demonstrated prime results of this combination in terms of treatment efficacy and generally mild toxicities.
7. Please provide a comprehensive table with all the included studies with data on G3 GEP- NET management. The authors should report apart from sample size, tumor origin and type of study, also patient outcomes in terms of ORR, DCR and PFS.
8. Please provide a comprehensive figure with a practical algorithm used at the Authors institution to address the management of G3 GEP-NETs based on contemporary literatture.
9. Along the same line, please discuss and refer to latest ESMO guidelines (Pavel et al. 2020, Annals of Oncology) on recommendations for treatment of G3 GEP-NET as well as SSR negative tumors in this setting, addressing the existing evidence on treatment selection and sequencing in the relevant sections of the present review.
10. Please consider to include a paragraph to better highlight the limitations of the present study including the scarcity of studies on this newly introduced entity of G3 NETs following the evolution of WHO grading classification in 2017, change of treatment paradigm for high grade NETs over time etc.
Author Response
Reviewer 1
The Authors present a review on the rare tumor entity of WD G3 GEP-NETs with the aims to present the latest data on the disease characteristics and evaluate its therapeutic management. In my view, the manuscript could be of interest to the clinical NET community as it summarizes well valuable information; therefore, I believe it would be of interest to the readership of Cancers. However, there are some major points that need to be addressed that could improve the quality of the paper. The manuscript would benefit from major revision, visualization of existing evidence in comprehensive tables and figures and clarification with respect to its limitations.
Reviewer’s comments.
- Please discuss briefly the suggested pathophysiological background of G3 NETs including high grade progression and provide a figure on the molecular pathogenesis of these tumors.
We thank the reviewer for his suggestion. We were not able to provide a figure to illustrate the pathophysiological background of NET G-3 but have added a paragraph in the Molecular biology and future directions section for the suggested pathophysiological background of NET G-3.
Modification in the revised manuscript, Molecular biology and future directions subsection, page 8:
“Indeed, NET G-3 seem to have common pathogenetic mechanisms with low and intermediate grades NEN, with key drivers of their molecular pathogenesis differing from those in NEC. As mentioned above, some researchers have suggested that PanNET G-3 might develop from an initial G-1 or G-2, whereas PanNEC could develop from ductal adenocarcinoma [58,62].”
- Please comply to PRISMA guidelines and report a search strategy from pertinent databases for the studies included in this review.
We thank the reviewer for his comment. Our review is not a systematic review addressing a specific question, but only a narrative review. This is why we have not followed the specific PRISMA guidelines and have not reported a search strategy in the paper. Similarly, we will not do a meta-analysis for the selected studies.
- With regards to functional and morphological imaging/radiomics please provide a table summarizing the evidence presented in the included studies discussed in the relevant sections.
We thank the reviewer for his suggestion. We have modified Table 2 and added data on functional imaging for the main available studies.
Table 2. Results from main studies evaluating grade 3 neurendocrine tumors (NET G−3) patients.
|
Study |
Velayoudom-Céphise et al. [6] 2013 |
Heetfeld et al. [3] 2015 |
Basturk et al. [7] 2015 |
Scoazec et al. [23] 2017 |
Hijioka et al. [4] 2017 |
Raj et al., [26] 2017 |
Kim et al.[21] 2020 |
|
Sample size |
12 |
37 |
19 |
21 |
21 |
16 |
8 |
|
Age (year) |
56 (median) |
52 (median) |
54 (median) |
NA |
63 (median) |
47 (mean) |
57 (median) |
|
Tumor location |
Pancreas, non-digestive sites |
Pancreas, rectum, stomach, small bowel |
Pancreas |
Pancreas, colon, rectum, stomach, small bowel |
Pancreas |
Pancreas |
Pancreas |
|
Metastatic state (%) |
100 |
62 |
67 |
NA |
71 |
69 |
100 |
|
Median Ki−67 (%) |
21 |
30 |
40 |
35 |
29 |
47 |
23 |
|
Positive SRI1 uptake -N2 of patients -Type of imaging -% of positivity |
-7/8
-Octreoscan®
-88 |
-21/24
-Not specified
-87,5 |
-
-
- |
-
-
- |
-
-
- |
-13/15
-Octreoscan®
-87 |
-
-
- |
|
Functional tumor |
3 (25%) |
5 (14%) |
1 (5%) |
- |
- |
8 (50%) |
0 (100%) |
|
ORR3 (%) |
- |
2/12 (17) |
- |
|
0/16 (0) |
- |
- |
|
Median OS4 (months) |
41 |
99
|
54 |
- |
42 |
52 |
87 |
|
Median PFS5 (months) |
- |
- |
- |
- |
- |
- |
16 |
|
DCR6 (%) |
- |
3/12 (33) |
- |
- |
6/16 (37,5) |
- |
- |
1SRI: somatostatin receptor imaging, 2N: number, 3ORR: overall response rate, 4OS: overall survival, 5PFS: progression free survival, 6DCR: disease control rate
- Along the same line, in the context of dual functional imaging with 68Ga-DOTA-TOC/TATE/NOC-PET and 18F-FDG-PET scan, please specify that PRRT has been proposed in this scenario under the condition of foci anatomical agreement of the gallium and glucose uptake lesions.
The rewiever is correct. We have added a sentence in the revised version of the manuscript.
Modification in the revised manuscript, PRRT subsection, page 11:
“PRRT has been proposed for high-grade NEN patients showing foci anatomical agreement of the SRI and glucose uptake lesions (little mismatch).”
- Mestier et al 2020 Neuroendocrinology, and Thomas et al. 2020 Cancers, provide data on TEMCAP treatment also in the subset of NET G3. Please consider including these references in the relevant discussion part.
Thank you for this suggestion. We have added a sentence and the 2 references in the revised version of the manuscript.
Modification in the revised manuscript, chemotherapy section, page 11:
“Similarly, other studies have found positive results with the CAPTEM regimen in small samples of NET G-3 [116–118].”
- With regards to PRRT, please consider to briefly discuss the novel therapeutic scheme of combining PRRT and chemotherapy in parallel for patients whose dual functional imaging scans show increased SSTR density and high glucose uptake. Several trials on agents such as 5-fluorouracil (5- FU), capecitabine (CAP) and temozolomide (TEM) have recently demonstrated prime results of this combination in terms of treatment efficacy and generally mild toxicities.
Thank you for this comment. We have added a sentence on this topic in the revised version of the manuscript.
Modification in the revised manuscript, PRRT subsection, page 11:
“Recently, the combination of PRRT and chemotherapy has been evaluated in advanced G-2 and G-3 NEN. As an example, one work found that CAPTEM combined with PRRT had significant activity with mild toxicities in a population including 8 G-3 NEN and showing positive dual tracer expression [118].
- Please provide a comprehensive table with all the included studies with data on G3 GEP- NET management. The authors should report apart from sample size, tumor origin and type of study, also patient outcomes in terms of ORR, DCR and PFS.
Thank you for this comment. We have modified Table 2 according to the reviewer’s suggestion and added data on ORR, DCR and PFS when it was available.
See Table 2 above (comment 3).
- Please provide a comprehensive figure with a practical algorithm used at the Authors institution to address the management of G3 GEP-NETs based on contemporary literatture.
Thank you for this suggestion. We have added a sentence in the conclusion section and created a new figure, Figure 3.
Modification in the revised manuscript, conclusion and perspectives section, page 12:
“Based on the available literature and on the 2020 European recommendations, we propose the following algorithm for the therapeutic management of digestive NET G-3 (Figure 3).”
Figure 3. Proposed treatment algorithm for the therapeutic management of G-3 well-differentiated neuroendocrine tumors (NET G-3).
CAPTEM: capecitabine + temozolomide, FOLFIRI: 5-fluorouracile + irinotecan, FOLFOX: 5-fluorouracile + oxaliplatine, XELOX: capecitabine + oxaliplatine, PRRT: Peptide receptor radionuclide therapy, SRI : somatostatin receptor imaging
- Along the same line, please discuss and refer to latest ESMO guidelines (Pavel et al. 2020, Annals of Oncology) on recommendations for treatment of G3 GEP-NET as well as SSR negative tumors in this setting, addressing the existing evidence on treatment selection and sequencing in the relevant sections of the present review.
Thank you for this suggestion. We have added the reference in the revised manuscript both in the treatment and conclusion sections.
Modification in the revised manuscript, conclusion and perspectives section, page 12:
“Based on the available literature and on the 2020 European recommendations, we propose the following algorithm for the therapeutic management of digestive NET G-3 [123] (Figure 3).”
- Please consider to include a paragraph to better highlight the limitations of the present study including the scarcity of studies on this newly introduced entity of G3 NETs following the evolution of WHO grading classification in 2017, change of treatment paradigm for high grade NETs over time etc.
Thank you for this suggestion. We have modified the Conclusion and Perspectives section by adding a paragraph in the revised version of the manuscript as follows:
Modification in the revised manuscript, conclusion and perspectives section, page 13:
“Overall, it is important to highlight that our work has limitations due to the scarcity of available studies for this newly introduced entity. Also, population samples are very small. Therefore, main conclusions on NET G-3 characteristics and specific treatment will need to be confirmed in future larger trials. In order to deal with recruitment difficulty due to the rarity of these tumors, international collaborations should be encouraged.”
Reviewer 2 Report
I commend the authors on highlighting the need for recognition of G3 NETs when diagnosing and treating patients with NENs. This is an evolving area of knowledge and the authors correctly identify the need for further research.
The structure of the paper 1) Epi 2) Imaging 3) Histo 4) Treatment sets up an outline that should be easy to follow, however the text is often confusing and the reader is unsure of how to interpret the data. There are many factual statements, but they are not tied together with consistency. This leaves the reader unsure of how each statement fits into the message that the author is trying to convene.
Please see the specific revisions below;
1) WHO 2017 changed the definitions for pancreatic NENs only, the digestive NENs classification change was not until 2019. Please reference appropriately
2) Line 77 - why is the NORDIC study included here if you are only speculating that some samples were NET G3, this does not add anything to the text
3) Line 87 - based on the studies you outlined NET G3's make up 10-20% of NEN G3 tumors (not the 1/3 you mention in line 88)
4) Line 95 - NET G-3 patients are more likely to have a functional tumor compared to NEC patients (14-25%). Where is the comparison? Does 14-25 represent NET or NEC? This is confusing
5) Last paragraph Clinical presentation and biomarkers - many studies with no interpretation as to the quality of the study or if they should be used for clinical decision making. What does the author want the reader to take away from this paragraph?
6) Imaging line 144-170 - this is very dense, it states the results of many trials, but leaves all interpretation up to the reader. For a review article, the author should be presenting data to the reader in a form that can assist the reader in decision making. The overwhelming feeling from reading this is that the reader should go read all the papers mentioned so they can gain an understanding of what the author is trying to convey.
7) Figure 1 - mislabelled there are two B's
8) Figure 2 - the difference between A and B is almost imperceptible on the images provided. ‘C’ is written as the copyright symbol. This figure does not add much to the readers understanding of NET G3. Is the imaging supposed to be different from G1/2 or NECs? A comparison may be more useful.
9) How much are non-functional imaging features used to diagnose the NEN vs NEC? This seems to be highlighted in the paper (e.g. Figure 2), but is not widely used in practice.
10) Table 2 - Velayoudom study, median Ki67 was 21% (this likely includes grade 2 NENs)?
11) Paragraph staring line 227 – Is the author suggesting using Ki67 to determine NET G3 vs NEC? This differs from the WHO guidelines which just uses degree of differentiation. It is still possible to have a NET with a Ki67 > 55. Is Ki67 essential for NEN diagnosis or just for grade of tumor (line234)? What is the author suggesting be used to measure Ki67 based on the literature? After reading the paragraph, the reader is still not clear if DIA, MC or ‘eyeballing’ should be used.
12) Is evolution of NENs proven? Or is there a possibility of sampling error? One retrospective study does not clarify this topic. Is the author suggesting multiple biopsies should be taken for all patients?
13) Molecular biology – this may be better communicated with a table
14) Line 284 – what is an ‘important initial tumor burden”?
15) Surgery and liver directed therapy - First paragraph – this gives this reader little guidance as to how they should approach NETG3’s surgically. It just reports several contradictory studies.
16) Line 356 – no ‘e’ on cisplatin in English
Author Response
Reviewer 2
I commend the authors on highlighting the need for recognition of G3 NETs when diagnosing and treating patients with NENs. This is an evolving area of knowledge and the authors correctly identify the need for further research.
The structure of the paper 1) Epi 2) Imaging 3) Histo 4) Treatment sets up an outline that should be easy to follow, however the text is often confusing and the reader is unsure of how to interpret the data. There are many factual statements, but they are not tied together with consistency. This leaves the reader unsure of how each statement fits into the message that the author is trying to convene.
Please see the specific revisions below;
- WHO 2017 changed the definitions for pancreatic NENs only, the digestive NENs classification change was not until 2019. Please reference appropriately
The reviewer is correct. We have made the change in the paper and added the appropriate reference.
Modification in the revised manuscript, introduction section, page 2:
“Well-differentiated grade-3 neuroendocrine tumors (NET G−3) have been introduced in the 2017 WHO classification for pancreatic lesions. This was later generalized to all digestive tumor sites [5]. »
“Table 1. The 2019 World Health Organization (WHO) Classification for Neuroendocrine Neoplasms (NEN) of the digestive tract.”
- Line 77 - why is the NORDIC study included here if you are only speculating that some samples were NET G3, this does not add anything to the text
Thank you for this comment, we have deleted the following sentences from the revised text:
“In the NORDIC study on 305 patients with GEP-NEN G-3 (only Ki−67 index > 20% tumors), we can speculate that some NET G−3 specimens were included since differentiation was not evaluated.”
“In the NORDIC study, PanNEN patients had higher rates of SRI uptake, in the context of possible undiagnosed NET G−3 tumors in this population (no information on differentiation).”
- Line 87 - based on the studies you outlined NET G3's make up 10-20% of NEN G3 tumors (not the 1/3 you mention in line 88)
The reviewer is correct. We have made the change in the revised version of the manuscript as follows:
Modification in the revised manuscript, tumor location and incidence sub-section, page 3:
“In summary, the incidence of NET G−3 is probably underestimated, but seems to account for 20% of the NEN G−3 population.”
- Line 95 - NET G-3 patients are more likely to have a functional tumor compared to NEC patients (14-25%). Where is the comparison? Does 14-25 represent NET or NEC? This is confusing
Thank you for this remark. We have changed the sentence in the revised version of the manuscript to make it less confusing, and added the data in Table 2.
Modification in the revised manuscript, clinical presentation and biomarkers sub-section, page 3:
“According to small available series, NET G−3 patients are more likely to have a functional tumor (5–50%) compared to NEC patients (0–6%) [3,6,7,26]”
Table 2. Results from main studies evaluating grade 3 neurendocrine tumors (NET G−3) patients.
|
Study |
Velayoudom-Céphise et al. [6] 2013 |
Heetfeld et al. [3] 2015 |
Basturk et al. [7] 2015 |
Scoazec et al. [23] 2017 |
Hijioka et al. [4] 2017 |
Raj et al., [26] 2017 |
Kim et al.[21] 2020 |
|
Sample size |
12 |
37 |
19 |
21 |
21 |
16 |
8 |
|
Age (year) |
56 (median) |
52 (median) |
54 (median) |
NA |
63 (median) |
47 (mean) |
57 (median) |
|
Tumor location |
Pancreas, non-digestive sites |
Pancreas, rectum, stomach, small bowel |
Pancreas |
Pancreas, colon, rectum, stomach, small bowel |
Pancreas |
Pancreas |
Pancreas |
|
Metastatic state (%) |
100 |
62 |
67 |
NA |
71 |
69 |
100 |
|
Median Ki−67 (%) |
21 |
30 |
40 |
35 |
29 |
47 |
23 |
|
Positive SRI1 uptake -N2 of patients -Type of imaging -% of positivity |
-7/8
-Octreoscan®
-88 |
-21/24
-Not specified
-87,5 |
-
-
- |
-
-
- |
-
-
- |
-13/15
-Octreoscan®
-87 |
-
-
- |
|
Functional tumor |
3 (25%) |
5 (14%) |
1 (5%) |
- |
- |
8 (50%) |
0 (100%) |
|
ORR3 (%) |
- |
2/12 (17) |
- |
|
0/16 (0) |
- |
- |
|
Median OS4 (months) |
41 |
99
|
54 |
- |
42 |
52 |
87 |
|
Median PFS5 (months) |
- |
- |
- |
- |
- |
- |
16 |
|
DCR6 (%) |
- |
3/12 (33) |
- |
- |
6/16 (37,5) |
- |
- |
1SRI: somatostatin receptor imaging, 2N: number, 3ORR: overall response rate, 4OS: overall survival, 5PFS: progression free survival, 6DCR: disease control rate
- Last paragraph Clinical presentation and biomarkers - many studies with no interpretation as to the quality of the study or if they should be used for clinical decision making. What does the author want the reader to take away from this paragraph?
Thank you for this comment. We have added a sentence at the end of the “clinical presentation and biomarkers paragraph” to give a take-home message for the reader:
Modification in the revised manuscript, clinical presentation and biomarkers sub-section, page 3:
“These preliminary results need confirmation by future studies. In total, the scarcity of available data makes it hard to draw firm conclusions on NET G-3 clinical presentation. Nevertheless, in case of diagnostic difficulty, a functional tumor is more in favor of NET G-3 diagnosis than NEC, especially if located in the pancreas. Furthermore, there are currently not enough data to recommend the dosage of any plasma biomarker in digestive NET G-3.”
- Imaging line 144-170 - this is very dense, it states the results of many trials, but leaves all interpretation up to the reader. For a review article, the author should be presenting data to the reader in a form that can assist the reader in decision making. The overwhelming feeling from reading this is that the reader should go read all the papers mentioned so they can gain an understanding of what the author is trying to convey.
We agree with the reviewer and have chosen to modify a part of this paragraph and also present the main data on functional imaging in Table 2.
Modification in the revised manuscript, functional imaging sub-section, page 4:
“In the small available series, most patients with NET G-3 showed positive SRI uptake (See Table 2) [3,6,26]. In Velayoudom-Céphise et al. [6] and Heetfeld et al. [3] studies, patients with NET G−3 showed significantly more positive SRI than those with NEC (p = 0.03 and 0,001 respectively). In a recent work, six NET G-3 lesions showed positive 68Ga-DOTATOC PET-CT uptake, and a negative correlation was found between Ki-67 index and SUVmax values in the whole NET population evaluated (r = -0.3, p = 0.018) [45]. Regarding 18F-FDG PET-CT, data are even more scarce (Table 2). In Velayoudom-Céphise et al. work, 25% of NET G-3 patients had a positive 18F-FDG PET-CT uptake [6]. In Heetfeld et al. work, only 12 patients of the NET G−3 group were evaluated and 9 had a positive 18F-FDG PET-CT uptake (75% of positivity), which was similar with NEC patients [3]. These first results indicate that NET G−3 are more likely to have a positive SRI and should benefit from this type of examination, but 18F-FDG PET-CT is not helpful for distinguishing between NET G−3 and NEC. Specific data is still lacking regarding the use of dual tracers in NET G-3, but some authors recommend that both SRI and 18F-FDG PET should indeed be performed given encouraging results with PRRT in this population [46].”
7) Figure 1 - mislabelled there are two B's
The reviewer is correct. We have made the change in the figure as follows, in the imaging section.
Figure 1. A 45-year-old woman presenting with a large lesion of the pancreatic head, with an initial diagnosis of grade 2 neuroendocrine tumor (NET G-2) after histopathologic analysis of biopsy specimens. Pre-therapeutic 68Ga-DOTATOC. Positron emission tomography/computed tomography (PET-CT) maximum intensity projection image (A) and axial fused PET-CT images (B, C) showed multiple liver lesions (B, arrow) and high uptake by pancreatic lesion (C). The 18F-FDG PET-CT (D, E, F) demonstrated high focal uptake only in the central region of the pancreatic tumor (F) and no pathological liver uptake (E), highlighting tumor heterogeneity. Finally, histopathologic analysis of surgical specimens revealed a NET G-3 with Ki-67 index of 22% and confirmed liver metastasis.
- Figure 2 - the difference between A and B is almost imperceptible on the images provided. ‘C’ is written as the copyright symbol. This figure does not add much to the readers understanding of NET G3. Is the imaging supposed to be different from G1/2 or NECs? A comparison may be more useful.
Thank you for this comment. First, we have removed the copyright symbol and apologize for the mistake.
Regarding Figure 2, we think it is important to keep it. Indeed, researchers have found that several morphologic features such as ill-defined margins, large tumor size, heterogeneous and poor to moderate enhancement, vascular involvement, upstream Wirsung duct dilatation, and distant metastases are less frequently observed in G-1 and G-2 NET by comparison with G-3 NEN. Therefore, non-functional imaging can help for the distinction of high-grade NEN and lower-grade NEN. Nevertheless, due to the rarity of NEC and G-3 NET, by comparison with the more frequent G-1 and G-2 NET, morphological criteria are mainly used in dedicated multidisciplinary meetings in centres with large output to predict invasiveness.
We have modified the revised version of the manuscript by adding the previous explanation and adding 3 references in the “morphological imaging and radiomics subsection. We have also modified the Figure 2 legend
Modification in the revised manuscript, Morphological imaging and radiomics sub-section, page 5:
“Researchers have found that several morphologic features such as ill-defined margins, large tumor size, heterogeneous and poor to moderate enhancement, vascular involvement, upstream Wirsung duct dilatation, and distant metastases are less frequently observed in G-1 G-2 NET by comparison with G-3 NEN [49–51]. Nevertheless, due to the rarity of NEC and G-3 NET morphological criteria are mainly used in dedicated multidisciplinary meetings in centers with large output to predict invasiveness.”
Modification in the revised manuscript, Morphological imaging and radiomics sub-section, Figure 2, page 6:
“(A),(E) The poor enhancement during arterial phase is in contrast with the classical hyperenhancement observed in well differentiated PanNET G-1 and G-2.”
- How much are non-functional imaging features used to diagnose the NEN vs NEC? This seems to be highlighted in the paper (e.g. Figure 2), but is not widely used in practice.
Thank you for this question. We have answered it in the previous comment (see above comment 8).
- Table 2 - Velayoudom study, median Ki67 was 21% (this likely includes grade 2 NENs)?
In Velayoudom-Céphise et al. work, only G-3 NEN patients were included and evaluated. To our knowledge, there were no G-2 NEN.
- Paragraph staring line 227 – Is the author suggesting using Ki67 to determine NET G3 vs NEC? This differs from the WHO guidelines which just uses degree of differentiation. It is still possible to have a NET with a Ki67 > 55. Is Ki67 essential for NEN diagnosis or just for grade of tumor (line234)? What is the author suggesting be used to measure Ki67 based on the literature? After reading the paragraph, the reader is still not clear if DIA, MC or ‘eyeballing’ should be used.
Thank you for this comment. We agree that this paragraph is a little confusing regarding interpretation. Overall, clinicians should follow the WHO guidelines for NEN G-3 classification, and MC is the gold standard for KI-67 index calculation.
We have modified the revised manuscript as follows:
Modification in the revised manuscript, morphology and Ki-67 index sub-section, page 8:
“Although no data on differentiation was available, the NORDIC study on 305 patients with GEP-NEN G-3 showed that patients with Ki-67 index < 55% had better OS and different treatment response [54]. Following these results, some authors have suggested that the 55% Ki-67 value could be the best cutoff to distinguish well-differentiated NEN G-3 from poorly differentiated NEN G-3 [55]. To this day, this has not been validated and clinicians should follow the WHO guidelines for NEN G-3 classification”
- Is evolution of NENs proven? Or is there a possibility of sampling error? One retrospective study does not clarify this topic. Is the author suggesting multiple biopsies should be taken for all patients?
We agree with the reviewer that this question remains unanswered but is a currently a hypothesis to explain the suggested pathophysiological background of G-3 NETs. There could be multiple explanations to a patient’s unexpected clinical evolution such as grade evolution or sampling error or even initial mixed grades. We do not suggest that multiple biopsies should be taken for all these patients but should be easily proposed to guide future treatment. We have modified the last paragraph of the “morphology and Ki-67 subsection” as follows:
Modification in the revised manuscript, morphology and Ki-67 index sub-section, page 8:
“Evolution of a well-differentiated NET to a high-grade NEN has been suggested by some authors. In a single-center retrospective study on 46 patients with 106 lesions of PanNET, increase in tumor grade occurred in 28 patients (63,6%) with the majority evolving from G-1/G-2 to G-3 [61]. On top of that, high progression correlated with worst survival [61]. In total, possible grade evolution remains a hypothesis. Other explanations such as sampling error or mixed tumors could also be considered in case of a patient’s unexpected clinical presentation. Therefore, multiple biopsies in one patient can sometimes be performed to guide future treatment”
13) Molecular biology – this may be better communicated with a table
Thank you for this suggestion. We have created a new Table, Table 3
Table 3. Main molecular alterations found in grade 3 neuroendocrine tumors (NET G−3) versus neuroendocrine carcinoma (NEC)
|
Molecular alterations (%) |
NET G-31 |
NEC2 |
|
Rb1 mutation |
0 |
67-75 |
|
KRAS mutation |
0 |
28-50 |
|
p53 mutation |
0 |
57-87 |
|
SMAD4 mutation |
0 |
5 |
|
Loss of DAXX/ATRX expression |
45-47 |
0 |
1NET G-3 : Grade 3 neuroendocrine tumor, 2NEC : neuroendocrine carcinoma
- Line 284 – what is an ‘important initial tumor burden”?
We meant “important initial tumor burden “in the sense of locally advanced tumors or when large resections were needed. We have made the change in the manuscript as follows:
Modification in the revised manuscript, subsection surgery and liver-directed therapies, page 9:
« Surgery can also be performed in well-differentiated NET after neoadjuvant therapeutic approach in patients with important initial tumor burden (locally advanced tumor or large resection needed), or in the presence of metastasis »
- Surgery and liver directed therapy - First paragraph – this gives this reader little guidance as to how they should approach NETG3’s surgically. It just reports several contradictory studies.
The reviewer is correct. We have made some changes in this paragraph to make it less confusing.
Modification in the revised manuscript, subsection surgery and liver-directed therapies, page 9:
“This is contradictory with current knowledge on NET G-3 prognosis and these results could be explained by the pooled evaluation of both localized and metastatic tumors [3,26,64,79]. Although surgery is recommended in the non-metastatic setting, we need to keep in mind that high-grade is associated with higher risk of recurrence and disease specific-death, as shown recently on a series of operated PanNEN [80]. In the metastatic setting, surgery can be a valid option for G-1 and G-2 NET. “
“Overall, although data on surgery in NET G−3 are scarce, it is still considered as the first valid option in the localized setting. It should be individually discussed for metastatic NET G-3 patients in the context of other available therapeutic approaches such as chemotherapy. Liver-directed therapies should be discussed individually and in case of important secretory syndrome.”
16) Line 356 – no ‘e’ on cisplatin in English
Thank you. We have made the correction in the revised version of the manuscript

Reviewer 3 Report
In this work, as part of a special issue on NET, the Authors review the current literature on the recently introduced category of well-diff NET G3.
The work help demarcate the difference between NET G3 and NEC G3 which is of help to the general medical readers and to the NET specialists.
The structure of the manuscript is set up in a classic manner and the content presentation is sound.
A brief reminder in the introduction regarding the concepts of tumor differentiation and grade might help take the reader into the topic more smoothly.
Author Response
In this work, as part of a special issue on NET, the Authors review the current literature on the recently introduced category of well-diff NET G3.
The work help demarcate the difference between NET G3 and NEC G3 which is of help to the general medical readers and to the NET specialists.
The structure of the manuscript is set up in a classic manner and the content presentation is sound.
A brief reminder in the introduction regarding the concepts of tumor differentiation and grade might help take the reader into the topic more smoothly.
We thank the reviewer for his comment. We have added a sentence in the introduction regarding the concepts of tumor differentiation and grade.
Modification in the revised manuscript, “introduction section”, page 2, line 49:
« The World Health Organization (WHO) grading system classifies neuroendocrine neoplasms (NEN) on tumor differentiation and tumor grade (proliferation rate). »